# Progress in Oral Microbiome Related to Oral and Systemic Diseases: An Update

**DOI:** 10.3390/diagnostics11071283

**Published:** 2021-07-16

**Authors:** Yeon-Hee Lee, Sang Wan Chung, Q-Schick Auh, Seung-Jae Hong, Yeon-Ah Lee, Junho Jung, Gi-Ja Lee, Hae Jeong Park, Seung-Il Shin, Ji-Youn Hong

**Affiliations:** 1Department of Orofacial Pain and Oral Medicine, Kyung Hee University Dental Hospital, #613 Hoegi-dong, Dongdaemun-gu, Seoul 02447, Korea; dental21@khu.ac.kr; 2Division of Rheumatology, Department of Internal Medicine, School of Medicine, Kyung Hee University, Dongdaemun-gu, Seoul 02447, Korea; wanyworld83@gmail.com (S.W.C.); hsj718@hanmail.net (S.-J.H.); aprildaum@hanmail.net (Y.-A.L.); 3Department of Oral and Maxillofacial Surgery, School of Dentistry, Kyung Hee University, Dongdaemun-gu, Seoul 02447, Korea; ssa204@khu.ac.kr; 4Department of Biomedical Engineering, Kyung Hee University, Dongdaemun-gu, Seoul 02447, Korea; gjlee@khu.ac.kr; 5Department of Pharmacology, School of Medicine, Kyung Hee University, Dongdaemun-gu, Seoul 02447, Korea; hjpark17@khu.ac.kr; 6Department of Periodontology, Periodontal-Implant Clinical Research Institute, School of Dentistry, Kyung Hee University, Dongdaemun-gu, Seoul 02447, Korea; shin.dmd@khu.ac.kr

**Keywords:** oral microbiome, oral cavity, dysbiosis, oral diseases, systemic diseases

## Abstract

The human oral microbiome refers to an ecological community of symbiotic and pathogenic microorganisms found in the oral cavity. The oral cavity is an environment that provides various biological niches, such as the teeth, tongue, and oral mucosa. The oral cavity is the gateway between the external environment and the human body, maintaining oral homeostasis, protecting the mouth, and preventing disease. On the flip side, the oral microbiome also plays an important role in the triggering, development, and progression of oral and systemic diseases. In recent years, disease diagnosis through the analysis of the human oral microbiome has been realized with the recent development of innovative detection technology and is overwhelmingly promising compared to the previous era. It has been found that patients with oral and systemic diseases have variations in their oral microbiome compared to normal subjects. This narrative review provides insight into the pathophysiological role that the oral microbiome plays in influencing oral and systemic diseases and furthers the knowledge related to the oral microbiome produced over the past 30 years. A wide range of updates were provided with the latest knowledge of the oral microbiome to help researchers and clinicians in both academic and clinical aspects. The microbial community information can be utilized in non-invasive diagnosis and can help to develop a new paradigm in precision medicine, which will benefit human health in the era of post-metagenomics.

## 1. Introduction

Human bodies are spatially shared by the oral microbiome and temporally serve as messengers or carriers of their genomes. That is, the human brain and nervous system recognize and control our body as an organism, but our body space is a biological recruitment unit that contains numerous microorganisms and their genomes. The number of microorganisms on the skin surface and inside the human body overwhelms the number of human cells [1]. The microbial community is referred to as the microbiome, a term used to signify the ecological community of commensal, symbiotic, and pathogenic microorganisms [1,2].

Over the past decade, various efforts have been made to identify and understand the role of the microbiome in human health and disease. With the advent of innovative genomics technologies, such as next-generation sequencing (NGS) and bioinformatic tools, which deviate from the conventional culture-based detection method, the understanding of the contribution of the human microbiome to health is deepening [3]. In addition, studies on the microbiomes in and on our bodies that form a functional organ fundamental to our health and physiology are still ongoing.

The microbiome refers to a community of microbial residents in the body. The term “microbiome” was coined by the 2001 Nobel Prize laureate Joshua Lederberg, who is credited for the Human Genome Project. The original meaning of the term is an ecosystem of symbiotic, commensal, and pathogenic microorganisms that reside in the human body [4]. The oral microbiome, oral microbiota, or oral microflora are microorganisms found in the human oral cavity, constituting bacteria, fungi, eukaryotes, and viruses (Figure 1). The genome is the genetic material of an organism, the complete set of DNA, including all of its genes. In terms of the genome, the oral microbiome is defined as the collective genome of microorganisms that reside in the oral cavity. The oral microbiome reaches homeostasis with regard to this composition, which is a dynamic balance in inter-bacterial and host–bacterial interactions [5].

As the oral cavity is ideal for microbiome residents, it is one of the most heavily colonized parts of the body. In addition, several distinct habitats within the oral cavity support complex and heterogeneous microbial communities. The interaction between oral microorganisms protects the human body against invasion and attack [6]. Conversely, an imbalance in the oral microbiome can contribute to oral diseases such as dental caries, periodontitis, oral mucosal diseases, and/or systemic diseases [7,8]. Furthermore, the relationship between the oral microbiome and the host is dynamic, and is influenced by many aspects of lifestyle, such as diet, tobacco consumption, stress, and systemic conditions, which can alter the composition and its properties and induce a state in which this finely tuned ecosystem is no longer in balance [4,9]. The oral microbiome is an important link between oral and general health.

We know that oral microbiomes are not colonized at random, but that our microbial residents have coevolved with us over millions of years. To address microbial homogeneity or divergence and maintain a harmonious state to maintain health and prevent disease, we must pay attention to our microbiomes and consider our body a superorganism. The purpose of this review is to inform oral and general healthcare practitioners on the current knowledge of the oral microbiome in health and disease, to review how molecular methods of microbial characterization have advanced our understanding, and to discuss potential implications for clinical practice. Moreover, with the recent development of technology for detecting and analyzing oral microbiomes, there is more detailed information on their existence, composition, and specific roles. For these purposes, we searched the literature via the PubMed and Google Scholar search engines, selected the top 169 articles that matched the theme for a period of 30 years prior to January 2021, and conducted an extensive and comprehensive narrative review.

## 2. The Uniqueness of the Oral Cavity as a Microbial Niche

The oral cavity has the most diverse microbiome in the digestive system. It is constantly exposed to both inhaled and ingested microbes, comprising more than 700 species of bacteria, fungi, viruses, archaea, and protozoans. Furthermore, saliva contains up to 10^9^ commensal bacteria per milliliter [10]. The oral cavity is an exceptionally complex habitat in which numerous microorganisms exist.

Within the oral cavity, distinct microenvironments exist, such as the hard surfaces of the teeth and epithelial surfaces of the mucosal membranes. The oral cavity consists of various habitats, such as the tongue, gingival sulcus, tonsils, hard palate, and soft palate, which provide a rich environment in which microorganisms can flourish [6]. Additionally, the supra-gingiva, sub-gingiva, interdental area, and tongue are distinct oral regions and have a unique microbiota composition [11,12,13,14,15]. These niches are primarily exposed to the fluid phase of the saliva or gingival crevicular fluid. This keeps the bacteria hydrated and serves as a medium for the transportation of nutrients to microorganisms [16]. As the initiation point of digestion, the oral cavity provides the nutrients necessary for the formation and maintenance of the oral microbiome.

Stable temperature and pH also provide an ideal environment for the growth of microorganisms. The human oral cavity is maintained at a relatively stable temperature of 35–37 °C. This temperature, without significant changes, is vital for the growth and survival of various microorganisms [17]. Saliva has a stable pH of 6.5–7, which is favorable for most species of bacteria [18]. The primary component involved in maintaining the neutral pH of the oral cavity is saliva; however, it has been found that different areas of the oral cavity have different pH [19]. The salivary microbiome has been shown to be a conglomerate of bacteria shed from oral surfaces, with the throat, tongue, and tonsils as the main sites of origin [20]. In addition, the oral microbiome composition differs between various habitats and has its own microbial identity, consisting of its unique microbial population [1].

## 3. The Salivary Microbiome in Health

The commensal microbiome plays an important role in the maintenance of oral and systemic health (Figure 2). The salivary microbiome has been shown to be individualized and temporarily stable in orally healthy individuals. However, limited information is available on the normal salivary microflora of healthy individuals [2,3]. The healthy human oral microbiome is predominantly composed of members of the phyla *Actinobacteria*, *Proteobacteria*, *Firmicutes*, *Bacteroidetes*, and *Fusobacter*, with *Spirochaetes* present in lower numbers [6]. The taxonomic profiles of spit, drool, and oral rinse samples were based on the proportion of bacterial sequences determined at the genus level [18]. The five major genera found in all three saliva fractions were *Streptococcus* (17.5%), *Prevotella* (15.5%), *Veillonella* (15.3%), *Neisseria* (12.7%), and *Haemophilus* (10%). The patterns of global diversity in any human microbiome based on analyses of partial 16S ribosomal RNA (rRNA) sequences from diverse locations around the world, the most frequent genus being *Streptococcus*, which accounted for 22.7% of the 101 bacterial genera, and 39 genera have not been previously described in the human oral cavity. Due to limitations in the ability to discriminate species using 16S ribosomal RNA analysis, the vast majority of studies have only identified oral bacteria at the genus level [4]. As there is significant variation in the structure and composition of the genomes of the same species, further research with both advanced sequencing and bioinformatic analysis is needed to characterize the oral microbiome at the strain level.

### 3.1. Oral Fungal and Protozoal Microbiota

There are very few reports on the role of the oral fungal and protozoal microbiota in health and disease [21]. However, other eukaryotic microbes, such as fungi and protozoa, are important non-bacterial components in the oral cavity. Fungi have been reported as members of healthy oral microbiota, where up to 101 species have been described, including *Candida* spp., followed by *Cladosporium*, *Aureobasidium*, *Saccharomyces*, *Aspergillus*, *Fusarium*, and *Cryptococcus* spp. [10,16]. Among the protozoa, *Entameba gingivalis* and *Trichomonas tenax* are the most common and are primarily saprophytic [22]. However, the detected *Candida* species are generally not associated with invasive human infections. In contrast, replacement with more virulent microbial species, including *Aspergillus* spp., *Fusarium* spp., and *Cryptococcus*, as part of an individual’s oral microbiome could potentially serve as a marker of increased risk of infection [23]. Archaea have also been detected, although they represent a minor proportion and are generally elevated in subjects with periodontitis [20].

### 3.2. Oral Viral Microbiota

Viruses are the most abundant infectious agents in different habitats, including other human body parts. The assemblage of viruses, the virome, has rarely been described in the human oral cavity [22,23] compared to its bacterial counterpart. Studies on oral viruses have generally focused on unstimulated saliva, dental plaque, or oral swabs from healthy individuals [24,25]. These viruses are predominantly bacteriophages and differ from the viromes described in other habitats. Approximately 10^8^ virus-like particles per milliliter of fluid from saliva swabs have been reported [24], and 10^7^ of them exist per milligram of dental plaque [25]. The most abundant eukaryotic viruses are herpesviruses (74%), particularly human herpesvirus (66%), followed by retroviruses (24%), and papillomaviruses (1.2%), with occasional counts of coronaviruses, poxviruses, and others (<1% of all reads) [26]. In general, viral infections focus on viruses that are found in or are transmitted via the oral cavity: norovirus, human papillomavirus, Epstein–Barr virus, herpes simplex virus, hepatitis C virus, and HIV. Determining the virulence factor homologs of oral viromes may support the notion that viruses can serve as reservoirs for pathogenic gene functions [27]. The presence of coronaviruses in saliva, particularly the 2019 novel coronavirus (SARS-CoV-2), has also been reported [28,29]. Although little is known about the role or constituents of viruses as members of the human microbiome, viruses could play a role as drivers of ecosystem diversity and are important contributors to the human oral microbiome in health and disease.

## 4. Effective and Rapid Detection of Oral Microbiota Diversity

In the past, analysis of the human oral microbiota has been hindered by the limitations of conventional methods. With conventional culture-based methods, many abundant oral microflora species remain unculturable. However, with the advancement of culture-independent approaches, including gel-based techniques, DNA microarrays, polymerase chain reaction (PCR)-based methods, and next-generation sequencing (NGS) technology, diverse unculturable flora have been identified [17,30]. The development of cost-effective high-throughput NGS technology has allowed researchers to easily obtain large amounts of DNA fingerprint data in a single instrumental run to decipher the complex oral microbial community in clinical samples such as saliva and subgingival plaque.

With the increasing clinical importance of oral pathogens, it is necessary to develop simple, rapid, and sensitive detection methods for point-of-care testing. The 2019 coronavirus disease (COVID-19) pandemic is currently the biggest challenge and a global health emergency [31]. Rapid and accurate diagnosis of COVID-19 is crucial for controlling the outbreak. Currently, reverse RT-PCR (rRT-PCR) is the gold standard for SARS-CoV-2 detection [32]. Saliva can have a significant role in human-to-human transmission, and salivary diagnostics may provide a comfortable and easy point-of-care platform for early and quick diagnosis of COVID-19 [33]. Detection of SARS-CoV-2 using saliva samples has been proposed as an alternative to standard swab diagnostic methods for the nose and pharynx [34].

NGS has represented the standard for studying the composition of microbial communities, allowing the differentiation of bacteria by sequencing the variable regions of the gene coding for the 16S ribosomal RNA (rRNA) (amplicon sequencing) [35,36]. Although 16S rRNA sequencing greatly improved our knowledge of the bacterial component of the oral microbiome, it only determines the presence or abundance of bacterial species. It usually does not provide sufficient information to resolve communities at the subspecies level, nor can it detect eukaryotic microorganisms [37]. With high-throughput NGS technologies, the genome of the entire community (metagenome) can be sequenced without the targeting step and without the bias of amplicon sequencing in PCR (shotgun whole metagenome sequencing). Furthermore, metagenomic analysis improves our knowledge of host–pathogen interactions by revealing the genes that potentially allow microbes to influence their hosts in unexpected ways [3]. Identification of viruses requires metagenomic sequencing, which refers to the direct sequencing of the total DNA extracted from a microbial community, due to the lack of the phylogenetic rRNA gene 16S.

To rapidly detect specific oral pathogens, methods based on the detection of specific bacterial DNA sequences have become invaluable in basic dental science and translational research [38]. In particular, multiplex real-time PCR is a sensitive method for detecting and quantifying a small number of bacteria in clinical samples. Real-time PCR offers a number of advantages over conventional PCR, including high sensitivity, improved accuracy, and the evaluation of data without post-PCR detection procedures [39,40].

## 5. Factors Influencing the Oral Microbiome

Changes in the oral and systemic environments can disrupt the normal symbiotic relationship between the host and its resident microorganisms and increase the risk of disease (Table 1). Various endogenous and exogenous factors, including smoking, alcohol consumption, socioeconomic status, antibiotic use, diet, and pregnancy, affect the oral microbiota (Figure 3) [41,42]. Disruption of the host–microbial mutualism, or dysbiosis, can occur because of significant changes in the oral environment or an individual’s lifestyle that favor the colonization of disease-associated microbiota [43].

Smoking is a significant factor that affects oral microbiota composition and orodental pathophysiology [44]. Toxic components in cigarettes can induce the loss of beneficial oral species and pathogen colonization, and eventually disease, directly or indirectly through immunosuppression, oxygen deprivation, or biofilm formation [45]. In smokers, poor commensals, such as *Streptococcus sanguinis* and *S. parasanguinis*, and abundant anaerobic microbiomes, such as *Fusobacterium nucleatum* and *F. naviforme*, were observed together with a high taxonomic diversity and richness, which was closely aligned with a disease-associated microbiota composition in clinically healthy individuals. Wu et al. [46] reported that the relative abundance of the phylum *Proteobacteria* was lower in current smokers than in never-smokers, with no difference between former and never-smokers. In addition, the *Capnocytophaga*, *Peptostreptococcus*, and *Leptotrichia* genera were depleted in current smokers compared with never-smokers, whereas the *Atopobium* and *Streptococcus* genera were enriched. Kumar et al. [47] reported differences between smokers and non-smokers in the formation of marginal and subgingival biofilms. In the biofilms of smokers, high taxonomic diversity and relatively unstable initial colonization were observed, with lower niche saturation than that observed in non-smokers. In particular, periodontal pathogens belonging to *Fusobacterium*, *Cardiobacterium*, *Synergistes*, and *Selenomonas* genera, as well as respiratory pathogens belonging to the genera *Haemophilus* and *Pseudomonas*, colonized the early biofilms of smokers [47].

Alterations in oral microbiota composition are expected during pregnancy. During pregnancy, women undergo complex and dramatic hormonal changes that affect the composition of the oral microbiota [48]. There have been reports suggesting an increased risk of periodontal disease during pregnancy and alteration of the composition of the oral microbiota [49,50,51]. *Porphyromonas gingivalis* in periodontal pockets has been linked to microbial invasion of the amniotic cavity in association with preterm labor [52]. Moreover, in a study that examined alterations in the oral microbiota between non-pregnant and pregnant states, increases in the total cultivable microbial counts and of periodontal pathogens such as *Porphyromonas gingivalis* and *Aggregatibacter actinomycetemcomitans* during pregnancy were observed in the gingival sulcus compared to that in non-pregnant women [53]. In particular, *Neisseria*, *Porphyromonas*, and *Treponema* were more abundant during pregnancy, while *Streptococcus* and *Veillonella* were less abundant [54].

Antimicrobial use has also been suggested as an important factor affecting the composition of oral microbiota. Amoxicillin treatment reduced species richness and diversity and shifted the relative abundance of 35 taxa [55]. They reported a substantial but incomplete recovery of the salivary microbiota composition from antibiotics approximately 3 weeks after the end of treatment. At the phylum level, the abundance of *Actinobacteria* was markedly decreased by amoxicillin, which was administered for approximately 10 days. This was recovered at the level of pretreatment approximately 3 weeks after amoxicillin treatment. In comparison, the abundance of *Proteobacteria* in saliva was increased about 3 weeks after the amoxicillin treatment compared to the pre-treatment saliva [55]. Raju et al. [56] reported the impact of antimicrobial use on saliva microbiota diversity and composition in preadolescents who systemically used amoxicillin, azithromycin amoxicillin-clavulanate, or phenoxymethylpenicillin. In addition, amoxicillin and amoxicillin-clavulanate potently decreased the abundance of *Rikenellaceae*. In children using azithromycin, a linear inverse association was observed between the use of azithromycin and the Shannon index [56].

Dietary intake also influences oral microbes. For instance, frequent sugar intake increases acid production, which dissolves tooth structure and increases the risk of dental caries via the fermentation of dietary carbohydrates by oral bacteria [57]. Dietary sugars and specific oral microbiota, including *Tannerella forsythia*, *Streptococcus sobrinus*, and *Eikenella corrodens*, can have a feedforward loop in patients with dental caries [13,43]. Subsequent acid production by repeated intake of high levels of carbohydrates led to sustained reductions in pH, along with the low buffering capacity of saliva. In turn, this can change the oral microbiota composition and upregulate aciduric species [58]. Aciduric species, including *Streptococcus mutans* and *Lactobacilli*, which are considered to be caries-associated microbiota, also produce acid under acidic conditions [59]. The change to acidic pH can lead to altered gene expression in sub-gingival bacteria, which favors the growth of pathogenic anaerobes such as *P. gingivalis*, which have an optimum pH for growth of approximately 7.5 [16]. Nutrition also affects periodontal bacteria and is considered a key modifiable factor for periodontitis [42].

There is controversy regarding whether alcohol consumption is a protective factor against or a risk factor for periodontitis. However, low socioeconomic status has been associated with periodontitis [60] and dental caries [61]. Alcohol consumption and socioeconomic status may also affect oral microbiome composition, considering reports on oral microbes in periodontitis and dental caries. Alterations in host immune competence can lead to an increase in the production of virulence factors, which can affect the community composition and meta-transcriptional landscape.

## 6. The Oral Microbiome within Common Oral Diseases

The oral microbiome associated with health is considered more general, whereas disease-associated microorganisms are specialists that elevate virulence potential, which is largely absent in healthy individuals. As a community shifts to dysbiosis, it ultimately facilitates the over-representation or overgrowth of microorganisms associated with dysbiosis.

### 6.1. Role of Oral Microbiome on Periodontitis

Periodontal disease is one of the most prevalent oral diseases worldwide. Periodontitis is characterized by the extension of inflammation into the supporting tissues of the teeth, causing the loss of attachment and bone. Progression of the disease may involve local, systemic, or environmental factors, and subsequent immune-inflammatory responses can occur in both hypo- and hyper-responsive manners [62]. It is a chronic non-communicable disease (NCD) with common risk factors with other NCDs, including cardiovascular disease, chronic obstructive pulmonary disease, diabetes, and cancer, which contribute to an increase in the global burden [63,64,65]. Gingivitis is an inflammation within the gingival tissue induced by the accumulation of bacterial deposits at the gingival crevice, which can be resolved by removing the deposits [66]. Despite the complex multifactorial features of the host response, the essential role of bacteria in the etiopathogenesis of gingivitis and periodontitis has long been identified [67,68,69].

Among the microbial habitats in the oral cavity, tooth surfaces harbor dental plaque biofilms in both supragingival and subgingival areas, and the bacterial deposits might vary depending on the anatomic factors such as surface topography, location, and shape of the teeth that affect bacterial stagnation and oral hygiene [70]. Individual variations in the biomass and composition of plaque are found between different sites of the same tooth or same sites of different teeth. More posterior locations and interdental space accumulated higher levels of plaque, whereas lingual surfaces showed limited debris compared to the buccal surface [71]. Further research and information on the spatial patterning of the oral microbiome associated with periodontal disease will further the understanding of the site-specific characteristics of the disease.

The crevicular epithelium and gingival crevice are oral microbial habitats critical for the initiation and development of gingivitis and periodontitis. The microbial community on the root surface can be protected from shear forces; the microenvironment is nourished by the gingival crevicular fluid (GCF), a serum-like exudate from the adjacent tissue, and a low redox potential to maintain anaerobic conditions. It has been estimated that more than 500 species exist in the subgingival plaque [72], and the current methods of high-throughput molecular technologies provide extended knowledge and understanding of the highly diverse microbial community in the oral cavity. The healthy subgingival microbiome is characterized by the dominant Gram-positive cocci and rods, *Actinomyces* spp. and *Streptococcus* spp., as early colonizers that co-aggregate and form early dental plaque [72]. The Gram-negative rod *Fusobacterium nucleatum* is the second most abundant species that acts as a secondary colonizer to bridge multiple bacteria as the plaque matures. Other Gram-negative species, such as *Veillonella* spp. and *Capnocytophaga* spp., are also important components of biofilms. Recently, Gram-positive *Rothia* spp. and *Corynebacteria* have emerged for their roles in spatial arrangements [73]. With the development of gingivitis, the subgingival microbiome shifts to an increase in Gram-negative bacteria, including *Prevotella* spp., *Selenomonas* spp., and *F. nucleatum* ss. *polymorphum* with a decrease in Gram-positive species and is involved in the elevation of inflammatory cytokines in GCF [74,75,76]. The total biomass of the bacteria, as well as compositional changes, increased by 3-log.

The periodontitis-associated subgingival microbiome can be described as the enrichment of diverse groups of Gram-negative species. In the classic fundamental study of Socransky et al., red complex bacteria comprising three species, *Porphyromonas gingivalis*, *Treponema denticola*, and *Tannerella forsythia*, are strongly associated with diseased sites and possess virulence factors such as gingipain, dentilisin, and PrtH, respectively, which showed high protease activity [6]. *Aggregatibacter actinomycetemcomitans* appears to be associated with aggressive periodontitis rather than chronic periodontitis, and it secretes leukotoxin to damage host cells. In terms of dysbiosis, *P. gingivalis* can subvert or avoid host immune components such as Toll-like receptors and complements, which triggers an imbalance in host–bacterial interactions and relative abundance of other bacteria compared with that in the healthy state [68]. Several keystone pathogens, including the red complex triad, orchestrated inflammatory disease by altering the microbiota associated with the disease state and through the optimization of the acquisition of nutrition from the host, and further facilitate the growth of pathobionts that stimulate host immune responses, resulting in bone loss [77,78] (Figure 4).

### 6.2. Role of Oral Microbiome and Dental Caries

Dental caries is characterized by the demineralization of susceptible dental hard tissues by acidic bacterial by-products from sugar metabolism that leads to cavitation [79] and a chronic continuous process from sub-clinical decay to dentinal involvement [80]. Dental caries results from an ecological imbalance towards acidogenic and aciduric bacterial shifts and environmental acidification within the dental biofilm, and can therefore be explained as a dysbiosis-associated disease [81]. Microbiota adapt to the reduced pH microenvironment along with the prolonged maturation of the biofilms; the cavitated lesions allow for advanced ecological niches [82].

Epidemiologic findings revealed a more frequent occurrence of dental caries in the occlusal and proximal surfaces of the first molars [83]. Pit and fissure structures or interdental spaces are anatomical factors that are protected from host defense mechanisms and avoid adequate removal of the plaque biofilm. Site-to-site variabilities in salivary film velocity and salivary clearance might affect the environmental selection and biogeography of the microbiome community, which in turn implies that the tooth surface is susceptible to demineralization [84].

Bacterial diversity was considerably different according to the disease conditions, as the supragingival plaque on the sound surface included 500–600 species, which was reduced to 200 in dentin caries and approximately 125 in non-cavitated enamel lesions [71]. In addition, the bacterial composition was variable among the different carious sites and lesions within the individual or among individuals [85,86]. Despite the complex microbial variability, the bacterial diversity of the caries-associated microbiome decreases due to the competitiveness of the microorganisms; the characterization of the cariogenic consortia should be further identified [87]. *Streptococcus mutans* are the initial colonizers of the supragingival biofilm, which produce water-insoluble glucans that promote bacterial adhesion and only comprise <1% of the total community [88]. In addition, *mutans streptococci*, especially *Streptococcus mutans* and *Streptococcus sobrinus*, were first identified as prominent pathogens because of their extensive acidogenic and aciduric properties [89,90]. The mutan-centric paradigm in caries-etiopathogenesis has been challenged by the identification of other acidogenic species, including *Bifidobacterium* [91], *Lactobacillus* [92] and *Scardovia wiggsiae* [93], which showed a strong association with caries. It was also revealed that a complex community associated with caries included other prominent species such as *Atopobium*, *Prevotella*, *Corynebacterium*, non-*mutans streptococci*, *Veillonella*, and *Capnocytophaga* [71,88]. These species produce weak organic acids after carbohydrate fermentation and decrease the local pH to demineralize the tooth tissues. Nevertheless, its potential role in triggering dysbiosis should not be underestimated.

### 6.3. Role of Oral Microbiome on Oral Lichen Planus

Oral lichen planus (OLP) is a chronic inflammatory mucocutaneous disease that mainly involves the oral mucosa. The etiology and pathogenesis of OLP are not clearly understood. However, OLP has been linked to multiple disease processes and agents, such as autoimmune diseases, allergic reactions to dental restorative materials, viral and bacterial infections, vaccinations, and medications [94]. So far, there has been little research on the oral microbiome in OLP. Bornstein et al. [95] reported higher bacterial counts of *C. sputigena*, *E. corrodens*, *L. crispatus*, *M. curtisii*, *N. mucosa*, *P. bivia*, *P. intermedia*, *S. agalactiae* and *S. haemolyticus* at the OLP lesion sites. Ertugrul et al. also reported that OLP patients had higher levels of infection with *A. actinomycetemcomitans*, *P. gingivalis*, *P. intermedia*, *T. forsythia*, and *T. denticola* than non-OLP patients [96]. Choi et al. found a decrease in *Streptococcus* and an increase in gingivitis/periodontitis-associated bacteria in OLP lesions [97]. Furthermore, they demonstrated that intracellular bacteria in the tissue and bacterial LPS might induce the production of T cell chemokines C-X-C motif chemokine ligand 10 and C-C motif chemokine ligand [98]. To date, the association between OLP and HCV viral infection appears to be dependent on geographical heterogeneity. This was first suggested by Mokni et al. in 1991, and Carrozzo et al. demonstrated a strong association between hepatitis C viral infection and OLP [99,100]. However, changes in the oral microbiome in patients with OLP remain unclear.

### 6.4. Role of the Oral Microbiome on Pre-Malignancy and Oral Cancers

Oral cancer is a cancer of the lips, mouth, or oropharynx. In an effort to elucidate the pathogenesis of oral cancers, oral microbiota has come into the spotlight and has been suggested to be involved through three possible mechanisms [101]. The chronic inflammatory responses provoked by bacteria could be responsible for this, since chronic inflammatory mediators facilitate cell proliferation, mutagenesis, and oncogene activation.

Anaerobic oral bacteria are known to cause chronic inflammatory processes in periodontal tissue by increasing interleukin-1β (IL-1β), IL-6, tumor necrosis factor-α, and matrix metalloproteinases MMP-8 and MMP-9 [102]. The bacterial effector proteins using type 3 or type 4 secretion systems might influence cell proliferation, cytoskeletal rearrangements, activation of NF-κB, and inhibition of cellular apoptosis [18,101]. In particular, *P. gingivalis* has been suggested to be anti-apoptotic. Additionally, the purinergic receptor P2X_7_ receptors activated by ATP are involved in cell death and apoptosis, and are affected by a nucleoside-diphosphate-kinase homolog, which is an ATP-consuming enzyme in *P. gingivalis* [103,104].

Carcinogenic substances such as acetaldehyde converted from ethanol, reactive oxygen species, reactive nitrogen species, and volatile sulfur compounds by bacteria might facilitate carcinogenic processes [105]. *S. gordonii*, *S. mitis*, *S. oralis*, *S. salivarius*, and *S. sanguinis* [91], and Candida can metabolize alcohol to acetaldehyde using the enzyme alcohol dehydrogenase [106]. ROS and RNS produced by peroxigenic oral bacteria, including *S. mitis*, *S. gordonii*, *S. sanguinis*, *S. oralis*, *L. fermentum*, and *Ljensenii*, have been identified in various oral cancers. VSCs have genotoxic effects and can cause genomic mutations [107]. *P. gingivalis*, *Pr. intermedia*, *A. actinomycetemcomitans*, and *F. nucleatum* are mainly responsible for the production of hydrogen sulfide (H_2_S), methyl mercaptan (CH_3_SH), and dimethyl sulfide ((CH_3_)_2_S) [108,109].

Recent data have indicated an association between human cytomegalovirus and carcinogenesis [110]. Human cytomegalovirus belongs to the beta subfamily of Herpesviridae and is frequently detected in periodontal tissues [111,112,113]. Proangiogenic and anti-apoptotic mechanisms and immunosuppressive effects demonstrate the oncogenic potential of human cytomegalovirus, and a non-lytic infection of the virus can also promote the oncogenic transformation of epithelial cells [110]. These data suggest that the virus may contribute to the oncogenic process of oral malignancies, although direct evidence is yet to be provided.

More studies investigating the pathophysiology of oral cancers are required to confirm the association between oral cancer and oral microbiota; however, evidence advocating for the association is accumulating, and oral cancer surfaces harbor significantly higher numbers of oral aerobes and anaerobes compared to the healthy mucosa surface [114]. Therefore, the detection and management of the oral microenvironment are essential for the control of oral malignancy.

## 7. Oral Microbiome and Systemic Diseases

The oral microbiome can also be a pathogenic factor in systemic diseases (Figure 5). The association between the oral microbiome and systemic diseases has been reported because the mouth is the entrance connecting the external environment and the inside of the body. A bidirectional relationship may exist between the oral microbial community and systemic diseases [9].

The oral microbiome and inflammatory molecules can invade the systemic organs in two primary ways: the bloodstream or the digestive tract. First, bloodstream invasion is possible because the periodontal pockets are anatomically close to the bloodstream. Second, by means of alimentary dissemination, the oral microbiome reaches the digestive tract. However, the precise mechanisms of invasion must be determined [8].

### 7.1. Autoimmune Disease

Dysbiosis of the oral microbiome plays a prominent role in several autoimmune diseases such as rheumatoid arthritis (RA), systemic lupus erythematosus (SLE), and primary Sjogren syndrome (SS) [115]. In autoimmune diseases, the most studied connection between the oral microbiome and disease is RA. The RA disease activity scores, DAS 28, were higher in patients with RA with more serum antibodies of *P. gingivalis*, an oral anaerobe involved in the development of periodontitis [116,117,118]. *P. gingivalis* produces gingipains and peptidylarginine deiminase, which enable protein citrullination, an important trigger for RA anticitrullinated peptide antibody [119,120]. In addition, many epidemiological studies have shown a correlation between RA and periodontitis [121,122]. In a recent meta-analysis of 21 studies, periodontitis was more frequent in patients with RA than in healthy controls, with a risk ratio of 1.13 [123].

Dysbiosis of the oral microbiome in SS has been reported in several studies [124,125,126,127,128,129,130]. A decrease in salivary secretion is the most important factor in oral dysbiosis in SS [128,129]. However, the impact of the oral microbiome on SS pathogenesis remains unclear. van der Meulen et al. reported higher *Firmicutes/Proteobacteria* ratios compared to those of healthy controls and higher abundances of 19 genera in SS patients. SLE is also associated with changes in the oral microbiome. In 2017, Corrêa et al. reported the influence of SLE on the subgingival microbiota, that the *Lachnospiraceae* family was increased compared to controls at healthy gingival sites, and the proportions of *Prevotella oulorum* and *Prevotella pleuritidis*, *Pseudomonas* spp., *Treponema maltophilum*, and *Actinomyces* found in healthy individuals [131]. Bacteria were elevated at periodontal sites compared with the controls. Periodontal disease is also associated with SLE, which increases the risk or severity of SLE [132,133].

### 7.2. Systemic Malignancies

In a recent longitudinal study of hospitalized cancer patients, increased variability in the oral microbiome was associated with adverse clinical outcomes [91]. In colorectal cancer (CRC), the role of microbiota in carcinogenesis and its clinical significance has been reported in many studies. The identification of the *Fusobacterium* genus in about 30% of CRC cases by 16S rRNA sequencing has initiated research on the impact of the oral microbiota in CRC [134,135,136]. *Fusobacterium nucleatum*, an oral commensal species, is more frequently identified in CRC than in colorectal adenoma, suggesting that *Fusobacterium* may contribute to later progression rather than an earlier stage along the colorectal adenoma-carcinoma sequence [137,138]. However, Kato et al. reported no association between *Fusobacterium* and CRC; instead, they observed associations with the genera *Lactobacillus* and *Rothia* by means of an NGS-based study [139]. They explained that *Lactobacillus* and *Rothia* were related to oral hygiene, and poor oral hygiene was related to CRC. Other microbes of oral microbiota, such as *Porphyromonas*, *Peptostreptococcus*, *Prevotella*, *Parvimonas*, and *Gemella*, are often associated with CRC. However, the carcinogenic potential and virulence factors of these genera are unknown [138].

Oral microbiota variation was associated with pancreatic cancer, and the combination of decreased *Neisseria elongata* with *Streptococcus mitis* was suggested to distinguish factors for pancreatic cancer [140]. Michaud et al. reported that the risk of pancreatic cancer was significantly increased in the presence of elevated serum antibodies against *P. gingivalis* [141]. Both *P. gingivalis* and *Aggregatibacter actinomycetemcomitans* have the potential to initiate Toll-like receptor pathways, which have been shown to be drivers of pancreatic carcinogenesis [142]. In 2016, a prospective large cohort study supported that *P. gingivalis* and *Aggregatibacter actinomycetemcomitans* may contribute to a higher risk of pancreatic cancer [143].

Esophageal cancer has also been reported to be related to *Tannerella forsythia* and *P. gingivalis* [144]. The study showed that the genus *Neisseria* was associated with a lower risk of esophageal cancer, as was the carotenoid biosynthesis pathway, to which a number of *Neisseria* species can potentially contribute.

Some studies have shown that periodontal diseases are associated with lung cancer risk [145,146,147]; however, it was difficult to analyze the association between oral microbiota and lung cancer because smoking, one of the biggest risk factors for lung cancer, can also affect the oral microbiota. In a small cross-sectional study, the abundance of *Capnocytophaga* and *Veillonella* was elevated, together with a reduced number of *Neisseria*, as found by means of 16S rRNA gene sequencing [148]. Thus, the current data are insufficient to conclude that oral microbial variations contribute to lung cancer.

### 7.3. Pregnancy Outcomes

Infection-related preterm birth is the leading cause of infant mortality and morbidity. Evidence indicates that ~40% of preterm births are vaginal and intrauterine infection-related, and ~50% are associated with intra-amniotic infections [149]. Therefore, an understanding of the origin of the offending bacteria and routes of invasion of the placenta and amniotic cavity is required. A review by Mendez et al. concluded that the most common intra-amniotic bacterial taxa were species associated with the vagina, although other species were commonly associated with the oral cavity, gastrointestinal tract, and respiratory tract [150].

Significant evidence supports an association between periodontal pathogenic bacteria, preterm birth, and preeclampsia. For example, *Fusobacterium nucleatum* is associated with adverse pregnancy complications [151], and *Porphyromonas gingivalis* has been detected in the amniotic fluid of pregnant women at risk for premature delivery and in the placentas of patients with preeclampsia [52,152].

### 7.4. Other Systemic Diseases

There is still conflict regarding the relationship between chronic inflammatory conditions, such as diabetes, and changes in the oral microbiome. In 2013, Chapple et al. reported that there is no compelling evidence that diabetes has a significant impact on the oral microbiota [153]. However, several studies have reported the impact of diabetes on changes in the oral microbiome. da Cruz et al. reported an increase in *P. gingivalis* and *Tannerella forsythia* in diabetes [154], and Ganesan et al. reported increased levels of *Capnocytophaga*, *Pseudomonas*, *Bergeyella*, *Sphingomonas*, *Corynebacterium*, *Propionibacterium*, and *Neisseria* in hyperglycemic individuals [155]. One NGS study reported that diabetes reduces *P. gingivalis*, *Tannerella forsythia*, and *Treponema denticola* [156]. To obtain a clearer conclusion, large-scale longitudinal studies are needed in the future.

The direct effect of oral bacteria on cardiovascular disease (CVD) is even less well-known, and the involvement of bacteria in atherosclerotic plaques in atherogenesis is not clear. However, since periodontitis and oral dysbiosis are related, oral dysbiosis is thought to be related to systemic inflammation. In a number of studies, the oral microbiota affected the outcome of CVD. The periodontal pathogen burden has been linked with acute coronary syndrome (ACS) and subclinical atherosclerosis [157,158,159], and Fak et al. reported that *Anaeroglobus* was more abundant in patients with symptomatic atherosclerosis than in controls [160]. Oral *Streptococcus sanguinis* moves from the oral cavity into the human bloodstream, colonizes the cardiac endothelium, and is a risk factor for infective endocarditis [161,162]. Lipopolysaccharide (LPS), endotoxin, and virulence factors of bacteria are considered a molecular link between the microbiome and cardiometabolic disorders [163]. Serum LPS activity correlates with the levels of the *P. gingivalis* antibody [164]. In addition, proteins secreted by *P. gingivalis*, such as gingipains, are implicated in their pathogenicity [165] and subsequently activate cytokine production [166]. There appear to be fewer studies using NGS techniques of oral cavity samples to determine the associations between the oral microbiome composition and cardiovascular disease.

Neurodegenerative disorders, such as Alzheimer’s disease [167] and Parkinson’s disease [168], have also been reported to be associated with the oral microbiome. The most complete study in this regard is the association between *P. gingivalis* and Alzheimer’s disease [167]. *P. gingivalis* infection in mice resulted in brain colonization, and gingipain proteases produced by *P. gingivalis* have neurotoxic characteristics. It has also been shown that typical oral species of the phylum Spirochaetes, including multiple species of the genus *Treponema*, often comprise amyloid plaques [169]. The potential mechanisms of their action by oral taxa make this another attractive area for investigation.

## 8. Conclusions

There is a dynamic interaction between the oral and systemic environments and the composition of the resident oral microbiome (Figure 6). Accumulated evidence suggests that the oral microbiome is individualized and relatively stable over time, as long as oral and general health are maintained. Substantial changes in key environmental parameters that affect microbial growth can disrupt the natural balance of the oral microbiome and select potentially pathogenic organisms. In addition, the presence of systemic diseases and oral microbiota seems to have a bidirectional effect. Thus, the oral microbiome reflects the oral and general health status of individuals. However, future studies are needed to determine whether changes in the oral microbiome precede clinical signs of disease, which would enable the use of the oral microbiome in the prediction of future disease risk. In our extensive and comprehensive narrative review, there may be a bias in the selection of papers based on keywords and for papers written in English. Accordingly, follow-up studies, including meta-analyses, are needed to reach a clearer conclusion. In addition, prospective longitudinal studies are urgently needed to reveal the full potential of using the oral microbiome in the field of precision medicine.

## Figures and Tables

**Figure 1 diagnostics-11-01283-f001:**
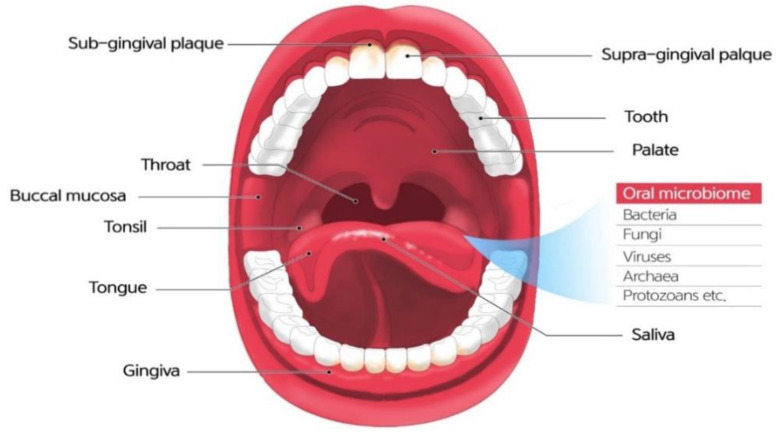
Oral cavity and the composition of oral microbiome.

**Figure 2 diagnostics-11-01283-f002:**
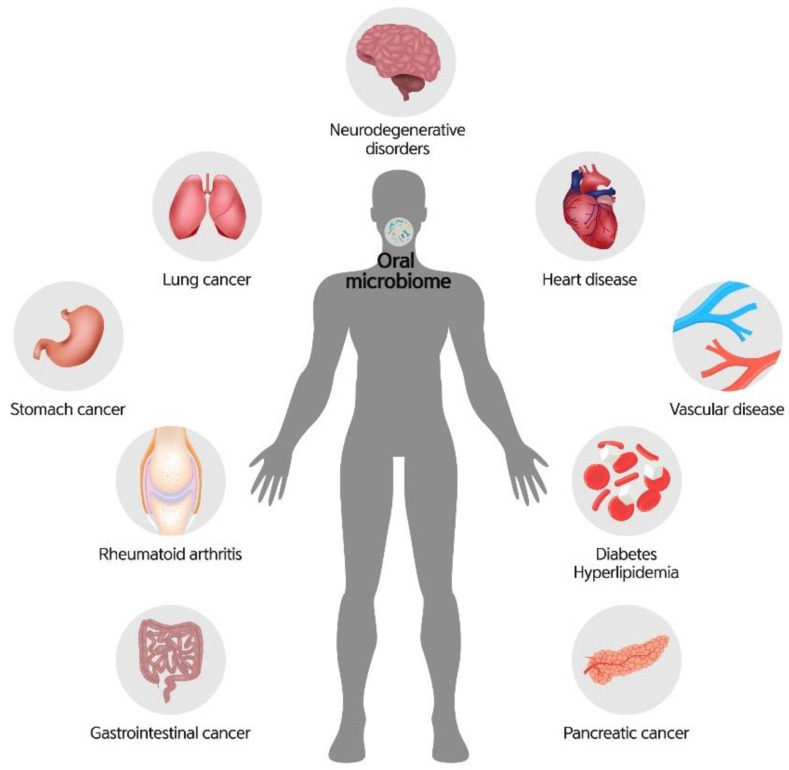
The oral microbiome is a crucial factor for systemic health.

**Figure 3 diagnostics-11-01283-f003:**
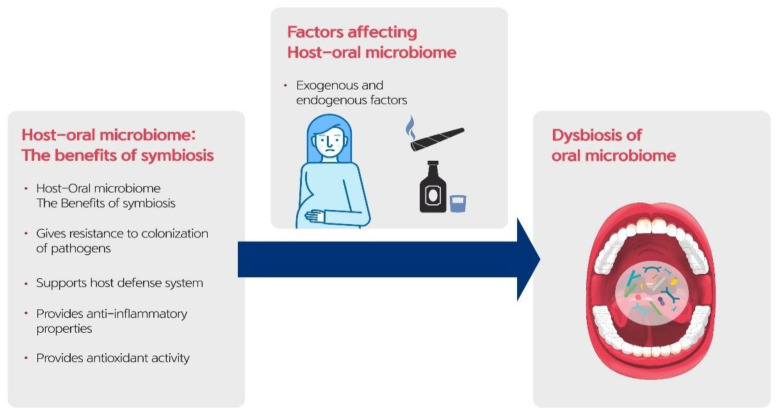
Various factors affecting the oral microbiome.

**Figure 4 diagnostics-11-01283-f004:**
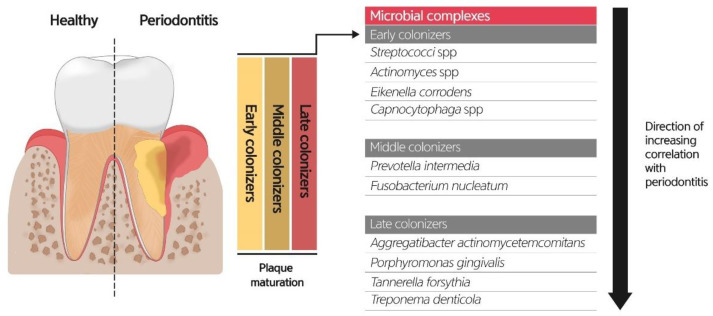
Bacterial colonizers related to periodontitis (adapted from [69,72,73]).

**Figure 5 diagnostics-11-01283-f005:**
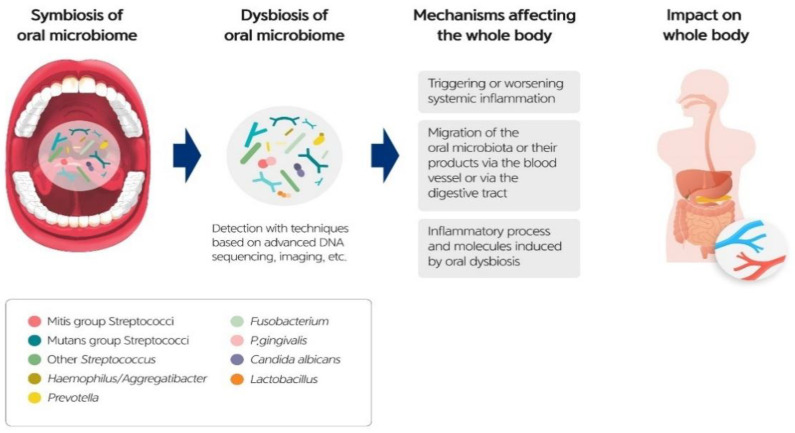
Exacerbation of oral microbial symbiosis to dysbiosis.

**Figure 6 diagnostics-11-01283-f006:**
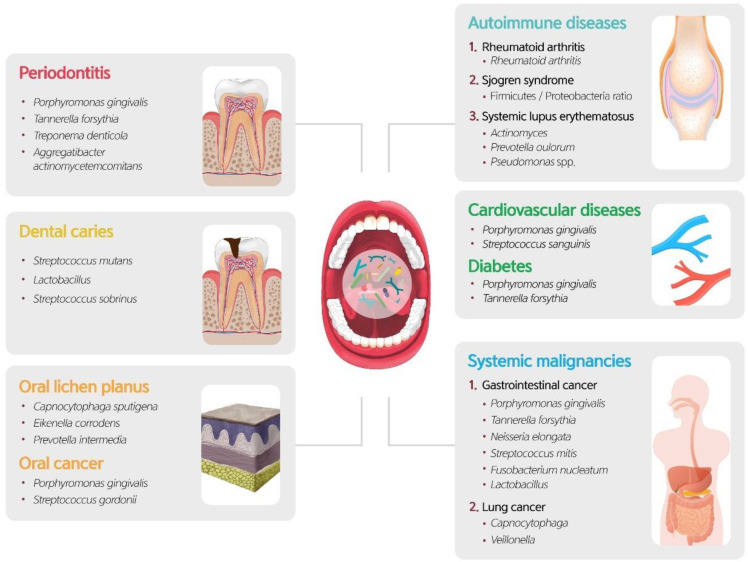
Types of systemic diseases related to dysbiosis of the oral microbiome.

**Table 1 diagnostics-11-01283-t001:** Various factors affecting the alteration of oral microbiome composition: analysis using 16S rRNA gene sequencing.

Factors	Samples	Major Finding
Smoking	Subgingival plaque	A high taxonomic diversity and richness in smokers
		Higher abundances of anaerobes in smokers: *Fusobacterium nucleatum*, *F. naviforme*, *Filifactor alocis*, *Dialister microaerophilus*, *Desulfobulbus* sp. *clone R004*, *Megasphaera sueciensis*, *M. geminatus*, *M. elsdenii*, *M. micronuciformis*, *Acinetobacter johnsonii*, *A. guillouiae*, *A. schindleri*, *A. baumannii*, *A. haemolyticus*, *Pseudomonas pseudoalcaligenes*, and *Pseudoramibacter alactolyticus*
	Saliva	Lower abundances of commensal microbes in smokers: *Streptococcus sanguinis*, *S. parasanguinis*, *S. oralis*, *Granulicatella elegans*, *G. adiacens*, *Actinomyces viscosus*, *A. israelii*, *A. dentalis*, *Neisseria subflava* and *Hemophilus parainfluenzae*
	Buccal mucosa	A lower taxonomic diversity in smokers
	Marginal and subgingival plaque and gingival crevicular fluid	High diversity in smokers Relatively unstable initial colonization of marginal and subgingival biofilms in smokers Periodontal pathogens belonging to the genera *Fusobacterium*, *Cardiobacterium*, *Synergistes*, and *Selenomonas*, as well as respiratory pathogens belonging to the genera *Haemophilus* and *Pseudomonas*, colonized the early biofilms of smokers and continued to persist over the observation period
	Oral wash samples	Lower abundance of the Proteobacteria phylum and genera belong to the Proteobacteria phylum in smokers Taxa not belonging to Proteobacteria: - increase: the *Atopobium* and *Streptococcus* genera in current smokers - decrease: the *Capnocytophaga*, *Peptostreptococcus* and *Leptotrichia* genera in current smokers
	Saliva	Increase: *Streptococcus sobrinus* and *Eubacterium brachy* in smokers
Antimicrobial agent	Saliva	Reduced species richness and diversity At the phylum level: - increase: Proteobacteria at post-treatment - decrease: Actinobacteria at the end of treatment
Antibiotics	Saliva	A linear inverse association between the use of azithromycin and Shannon index Amoxicillin and amoxicillin-clavulanate use was associated with the largest decrease in abundance of *Rikenellaceae* Phenoxymethylpenicillin were associated with a decrease in *Paludibacter*
Pregnancy	Supragingival plaques	Significantly higher Shannon diversity in pregnant women More abundant: *Neisseria*, *Porphyromonas*, and *Treponema* in pregnant women Less abundant: *Streptococcus* and *Veillonella* in pregnant women

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
