# Peer review of "Progress in Oral Microbiome Related to Oral and Systemic Diseases: An Update"

_diagnostics, 2021, doi:10.3390/diagnostics11071283_

Round 1

Reviewer 1 Report

The manuscript is very well return and organized and comprehensive and addresses the subject in totality, it is however extremely long and can benefit from reduction in length.  Section 6 may be expanded to cover congenitally related disorders among the pathologies covered.  Additionally aside from the oral pre-dysplasia and oral carcinogenesis the authors are encouraged to also cover cytomegalovirus related malignancies including hematologic and expand a bit more in the mechanism related to microorganisms

Author Response

Response:

The authors sincerely thank you for your accurate and valuable comments. We did our best to faithfully reflect the opinions of the three reviewers and went through several discussions. Modified parts of manuscript are marked in red. The authors would like to thank the reviewers and the journal once again for giving us a positive response and providing a potential opportunity to contribute to your wonderful journal. Thank you very much.

Comments and Suggestions for Authors

The manuscript is very well return and organized and comprehensive and addresses the subject in totality, it is however extremely long and can benefit from reduction in length.  Section 6 may be expanded to cover congenitally related disorders among the pathologies covered.  Additionally aside from the oral pre-dysplasia and oral carcinogenesis the authors are encouraged to also cover cytomegalovirus related malignancies including hematologic and expand a bit more in the mechanism related to microorganisms.

Response: The authors sincerely thank you for your accurate and valuable comments. We have maintained the essential elements of the manuscript according to your comments, but have deleted unnecessary parts.

In addition, we respect your opinion and have incorporated the content of cytomegalovirus related malignancies into our paper. Thank you.

Reviewer 2 Report

Dear authors,

The article “Progress in Oral Microbiome related to Oral and Systemic Dis-2 eases: An Update” is interesting and summarized briefly the link between the oral microbiota and oral and systemic diseases.

The authors explained that they did a systematic review so the article should be revised to be in accordance with PRISMA for systematic reviews. The sections should be reorganized. The number of figures is too important and a lot of figures are not necessary because they contain too little information or misunderstanding. Please consider my comments:

Introduction section

  1. Add some other recent references such as PMID: 32102216; DOI: 10.1007/978-3-030-42990-4_10; PMID: 31600905…
  2. “top 147 articles » This is not clear. Please explain how you selected these articles: keywords, type of articles,… Did the selection was made by one researcher?
  3. “massive systematic review” Clarify “massive”. If you did a systematic review, you should follow PRISMA and revised your manuscript in accordance with PRISMA.

Section “What is the oral microbiome?

  1. This section is a definition of the microbiome and should be reduced and replace in the introduction section.

Section “The uniqueness of the oral cavity as a microbial niche”

  1. This section should be entirely revised because it lacks a lot of information. It focused more on the saliva microbiota than on the oral microbiota. You should describe the supragingival microbiota (PMID: 30482830), the subgingival microbiota (PMID: 25814980), the interdental microbiota (PMID: 31491909, PMID: 27313576), the tongue microbiota (PMID: 30111628),…
  2. “Stable temperature and pH also provide an ideal environment for the growth of microorganisms. The human oral cavity is maintained at a relatively stable temperature of 35–37  C. This temperature, without significant changes, is vital for the growth and survival of various microorganisms. Saliva has a stable pH of 6.5–7, which is favorable for most species of bacteria.” Add references

Section “The salivary microbiome in health”

  1. “However, limited information is available on the normal microflora of healthy individuals [2, 13].” Add “salivary normal microflora”
  2. “There are very few reports on the fungal microbiome and other microorganisms.” Precise the type of microbiota you referred to. Is it “oral microflora”? “salivary”…? And add reference
  3. “We should recognize the critical contribution of the mycobiome as a trigger for immune responses in diseases. ” This sentence is a point of view and should be deleted

Section “Oral viral microbiota”

  1. Please add the presence of coronaviruses in the saliva and particularly SARS-CoV-2 (PMID: 34066046, PMID: 32047895, PMID: 32449329)

Section “Effective and rapid detection of oral microbiota diversity”

  1. This section focused on the technology to analyze, describe and quantify the microbiota. For me, this section should be deleted or summarize in 2 or 3 sentences and added in the introduction

Section “Factors influencing the oral microbiome”

  1. “Changes in oral and systemic environments can disrupt the normal symbiotic relationship between the host and its resident microorganisms and increase the risk of disease. Various endogenous and exogenous factors, including smoking, alcohol consumption, socioeconomic status, antibiotic use, diet, and pregnancy affect oral microbiota (Figure 3).” References should be added for each factor (PMID: 31307469, PMID: 33430519…)
  2. In this section, the 3rd paragraph focused on the pregnancy. Why have chosen the pregnancy because it is known that other hormonal changes can also affect the oral microbiota (PMID: 33816334,
  3. In the paragraph analyzing the diet, please add the impact of diet on periodontal bacteria also (PMID: 33430519) and also add the potential importance of S. sobrinus, E corrodens and T. forsythia (PMID: 31491909)
  4. Reference 33 is very old. Please replace by one more recent

Section “6-1 Role of oral microbiome on Periodontitis”

  1. In this section, you should also describe the role of the interdental microbiota in the initiation of periodontal lesions

Section “6-2 Role of oral microbiome and dental caries”

  1. In this section, you should also describe the role of the interdental microbiota in the initiation of carious lesions

Section “7. Oral microbiome and systemic diseases”

  1. In the introduction of this section, you should describe how the oral microbiome and systemic diseases are linked via the dissemination of oral bacteria, via the inflammatory process…(PMID: 31600905).
  2. You should also discuss the impact of the dysbiosis of oral microbiota on adverse pregnancy outcomes

Figures

  1. Figure 1 should be deleted because there is no interesting information
  2. Revise figure 2 because “lung cancer” appears 3 times and the pictures don’t correspond to the lung. Same remarks for “cardiovascular diseases”
  3. Revise figure 3 because the arrows do not show a logical link. The “host-oral microbiome- The benefits of symbiosis” doesn’t induce “Influencing factor”
  4. Figure 5 should be deleted or revise because they lack a lot of middle colonizers as described by Socransky complexes. The reference to Socransky complexes should be added
  5. Figure 5 The dysbiosis of the oral microbiota was not only determined by sequencing. You should delete “detection with techniques based…”. Moreover, the “mechanisms affecting the whole body” are not clear because the “sustained microbiome dysbiosis” is not a mechanism but is a consequence of the (i) migration of the oral microbiota or their products via the blood vessel or via the digestive tract… or by the (ii) inflammatory process and molecules induced by oral dysbiosis…
  6. Figure 6 should be completed because they lack some bacteria such as Streptococcus sanguinis that is linked to endocarditis (PMID: 29882414

Author Response

Response:

The authors sincerely thank you for your accurate and valuable comments. We did our best to faithfully reflect the opinions of the three reviewers and went through several discussions. Modified parts of manuscript are marked in red. The authors would like to thank the reviewers and the journal once again for giving us a positive response and providing a potential opportunity to contribute to your wonderful journal. Thank you very much.

Best regards,

Dear authors,

The article “Progress in Oral Microbiome related to Oral and Systemic Dis-2 eases: An Update” is interesting and summarized briefly the link between the oral microbiota and oral and systemic diseases.

The authors explained that they did a systematic review so the article should be revised to be in accordance with PRISMA for systematic reviews. The sections should be reorganized. The number of figures is too important and a lot of figures are not necessary because they contain too little information or misunderstanding. Please consider my comments:

Response: Thank you very much for your valuable comments and suggestions. We declare that we have done everything we can to reflect your attentive and accurate comments over the past 10 days. We revised the terminology of 'systematic review' of Introduction to 'narrative review' by combining the opinions you and Reviewer 3 gave. Thank you.

Introduction section

  1. Add some other recent references such as PMID: 32102216; DOI: 10.1007/978-3-030-42990-4_10; PMID: 31600905…

Response: We've added your latest references to the Introduction. Thank you so much.

  1. “top 147 articles » This is not clear. Please explain how you selected these articles: keywords, type of articles,… Did the selection was made by one researcher?

Response: All authors participated in writing a certain part of the manuscript according to their major and specialty. Of course, the first and corresponding author Prof. Lee (Yeon-Hee Lee) made the biggest contribution throughout the study. Our thesis is an extensive narrative review, combining the parts written by each author for a sub-topic, and we did not find references only with special key words. In the synthesis of Manuscript, the role played by Prof. Lee was the most decisive and important. During the revision, all authors provided their opinions and experiences, and we went through several discussions. Thank you.

  1. “massive systematic review” Clarify “massive”. If you did a systematic review, you should follow PRISMA and revised your manuscript in accordance with PRISMA.

Response: Thank you very much for your precise comments and suggestions. We revised the terminology of 'systematic review' of Discussion to 'narrative review' by combining the opinions you and Reviewer 3 gave.

Of course, we are well aware of the purpose of the review article that follows, and we have worked hard to achieve these goals.

  • Provide a comprehensive foundation on a topic
  • Explain the current state of knowledge
  • Identify gaps in existing studies for potential future research
  • Highlight the main methodologies and research techniques

Thank you very much.

Section “What is the oral microbiome?

  1. This section is a definition of the microbiome and should be reduced and replace in the introduction section.

Response: Thank you for your comments and suggestions. Based on your opinion, we shorten the description of the oral microbiome section and move the revised content from the main body to the Introduction section.

Section “The uniqueness of the oral cavity as a microbial niche”

  1. This section should be entirely revised because it lacks a lot of information. It focused more on the saliva microbiota than on the oral microbiota. You should describe the supragingival microbiota (PMID: 30482830), the subgingival microbiota (PMID: 25814980), the interdental microbiota (PMID: 31491909, PMID: 27313576), the tongue microbiota (PMID: 30111628),…

Response: The authors sincerely thank you for your attentive and meticulous comments and suggestions. We found all the latest references you gave us and reflected them in the manuscript. Thank you.

  1. “Stable temperature and pH also provide an ideal environment for the growth of microorganisms. The human oral cavity is maintained at a relatively stable temperature of 35–37  C. This temperature, without significant changes, is vital for the growth and survival of various microorganisms. Saliva has a stable pH of 6.5–7, which is favorable for most species of bacteria.” Add references

Response: The authors sincerely thank you for your warm, precise and meticulous comments and suggestions. We found and added references to the places you pointed out. Thank you.

Section “The salivary microbiome in health”

  1. “However, limited information is available on the normal microflora of healthy individuals [2, 13].” Add “salivary normal microflora”.

Response: The authors sincerely thank you for your meticulous comments. We added the word 'salivary' to the point you pointed out. Thank you.

  1. “There are very few reports on the fungal microbiome and other microorganisms.” Precise the type of microbiota you referred to. Is it “oral microflora”? “salivary”…? And add reference

Response: The authors sincerely thank you for your comments. We added the word 'oral' to the point you pointed out. Furthermore, we added the reference, properly. Thank you.

  1. “We should recognize the critical contribution of the mycobiome as a trigger for immune responses in diseases. ” This sentence is a point of view and should be deleted

 Response: We fully respect your opinion, and we have removed that sentence.

Section “Oral viral microbiota”

  1. Please add the presence of coronaviruses in the saliva and particularly SARS-CoV-2 (PMID: 34066046, PMID: 32047895, PMID: 32449329)

Response: The authors sincerely appreciate your constructive and accurate comments and suggestions. Based on your comments, we added sentences and references. Thank you very much.

Section “Effective and rapid detection of oral microbiota diversity”

  1. This section focused on the technology to analyze, describe and quantify the microbiota. For me, this section should be deleted or summarize in 2 or 3 sentences and added in the introduction

Response: Thank you for your comments, and we consider this part to be what sets our paper apart from other papers. Therefore, rather than completely deleting the part you pointed out, we added a sentence introducing this part to the introduction. We ask for your broad understanding.

Section “Factors influencing the oral microbiome”

  1. “Changes in oral and systemic environments can disrupt the normal symbiotic relationship between the host and its resident microorganisms and increase the risk of disease. Various endogenous and exogenous factors, including smoking, alcohol consumption, socioeconomic status, antibiotic use, diet, and pregnancy affect oral microbiota (Figure 3).” References should be added for each factor (PMID: 31307469, PMID: 33430519…)

Response: Thank you very much for your attentive comments. We added references to that section.

  1. In this section, the 3rd paragraph focused on the pregnancy. Why have chosen the pregnancy because it is known that other hormonal changes can also affect the oral microbiota (PMID: 33816334,

Response: Thank you very much for your wise and valuable comments. Pregnant women are more likely to develop periodontitis, and we wanted to provide information on whether their oral microbiota composition is different. To help readers understand, we have added your point to the manuscript.

  1. In the paragraph analyzing the diet, please add the impact of diet on periodontal bacteria also (PMID: 33430519) and also add the potential importance of S. sobrinus, E corrodens and T. forsythia (PMID: 31491909)

Response: Thank you for your comments and suggestions. We have updated and added all of your recommendations. Thank you.

  1. Reference 33 is very old. Please replace by one more recent

Response: Thank you very much for your attentive comments. Following your comments, we replaced reference #33 with a more up-to-date one.

Section “6-1 Role of oral microbiome on Periodontitis”

  1. In this section, you should also describe the role of the interdental microbiota in the initiation of periodontal lesions

Response: Thank you for the important comment. Interdental spaces are known to be the unique ecological habitats where daily oral hygiene procedures are inadequate to be reached and accumulation of bacterial deposits may occur in higher levels compared to other surfaces. In addition to the approximal tooth sites, anatomic factors including surface topography (pit, fissure and irregularities on hard dental tissues), location (anterior vs. posterior teeth) and shape of the teeth also affect the bacterial stagnation and the patterns of microbial deposition may be linked on a site-specific characteristic of periodontal disease. There are still limited data about the microbial profiles and spatial patterns contributing to the initiation and/or progression of the periodontal disease, which should be determined through the future works. These contents were added in the manuscript with references.

Section “6-2 Role of oral microbiome and dental caries”

  1. In this section, you should also describe the role of the interdental microbiota in the initiation of carious lesions

Response: Thank you for the comment. As described above, interdental spaces are known to be the unique ecological habitats where daily oral hygiene procedures are inadequate to be reached and accumulation of bacterial deposits may occur. Previous epidemiologic studies have reported that dental caries showed higher frequency in the occlusal and proximal sites of the first molars. Anatomic factors of occlusal irregularities such as pits and fissures and proximal surfaces provide hard-to-reach sites for disruption of the plaque biofilm. In addition, salivary film velocity and salivary clearance varies site-to-site that might affect the microenvironment at the tooth surface and the growth of the microbial community. Despite the limited data in spatial patterning in the distribution of caries and microbial profiles, studies on the biogeography of the microbiome community might help understand the tooth surfaces with high or low susceptibility to demineralization. These contents were added in the manuscript with references.

Section “7. Oral microbiome and systemic diseases”

  1. In the introduction of this section, you should describe how the oral microbiome and systemic diseases are linked via the dissemination of oral bacteria, via the inflammatory process…(PMID: 31600905).

Response: Thank you very much for your comments. As you commented, we amended the sentences in the introduction of this section as below;

Oral microbiome and inflammatory molecules can invade the systemic organs in two main ways: the bloodstream or the digestive tract. First, bloodstream invasion is possible because anatomically, the periodontal pockets are close to the bloodstream. Second, by alimentary dissemination, oral microbiome, reach the digestive tract. However, the precise mechanism of invasion must be determined.

  1. You should also discuss the impact of the dysbiosis of oral microbiota on adverse pregnancy outcomes·

Response: Thank you for your comments and suggestions. As you recommended, we added the section of oral microbiota on adverse pregnancy outcomes.

6-4         Other systemic diseases

Figures

  1. Figure 1 should be deleted because there is no interesting information

Response: I respect your opinion. However, within our group, as we proceeded with this study, we became clear about the definitions of terms such as oral microbiome and oral microbiota. Many people know that the oral microbiome usually contains only bacteria, so this figure can enhance the reader's understanding. Since you are a great expert in this field, Figure 1 can be taken for granted. Thank you for your broad understanding.

  1. Revise figure 2 because “lung cancer” appears 3 times and the pictures don’t correspond to the lung. Same remarks for “cardiovascular diseases”

Response: In fact, the authors really appreciate and are surprised by your meticulous review. We have corrected all the points you pointed out in Figure 2. Thank you.

  1. Revise figure 3 because the arrows do not show a logical link. The “host-oral microbiome- The benefits of symbiosis” doesn’t induce “Influencing factor”

Response: We modified the relationship and schematic diagram in Figure 3, as you pointed out. Thank you sincerely.

  1. Figure 5 should be deleted or revise because they lack a lot of middle colonizers as described by Socransky complexes. The reference to Socransky complexes should be added

Response:  In response to your comments, we have added 'Socransky complexes' and a reference to the title of Figure 4.

  1. Figure 5 The dysbiosis of the oral microbiota was not only determined by sequencing. You should delete “detection with techniques based…”. Moreover, the “mechanisms affecting the whole body” are not clear because the “sustained microbiome dysbiosis” is not a mechanism but is a consequence of the (i) migration of the oral microbiota or their products via the blood vessel or via the digestive tract… or by the (ii) inflammatory process and molecules induced by oral dysbiosis…

Response: We have accepted all of your suggestions and have fully reflected them in Figure 5. Sincerely thank you for your wise advice.

  1. Figure 6 should be completed because they lack some bacteria such as Streptococcus sanguinis that is linked to endocarditis (PMID: 29882414)

Response: We fully agree with you and have added Streptococcus sanguinis to Figure 6. In addition, we have added this content to the manuscript along with the reference. Our paper has progressed one step further thanks to your comments and suggestions. Thank you very much.

Submission Date

21 June 2021

Date of this review

30 Jun 2021 23:35:59

Reviewer 3 Report

The paper is a revision on the role of the oral microbiome as an indicator for systemic pathologies. The review is an extensive and cohomprehensive summary which sounds scientifically and is properly presented. However, a fundamental aspect have to be clarified: the paper is inteded to be a narrative review, but the authors used the terminology "in our systematic review", I please the authors to check for this aspect since the systematic review is a kind of paper which is so far from the author's one.

The figures have to be inserted in the manuscript next to the citation point and not at the end of the paper, please check for authors guidelines.

Please, reference section have to be checked for discrepancies according to the authors guidelines.

Author Response

Response:

The authors sincerely thank you for your accurate and valuable comments. We did our best to faithfully reflect the opinions of the three reviewers and went through several discussions. Modified parts of manuscript are marked in red. The authors would like to thank the reviewers and the journal once again for giving us a positive response and providing a potential opportunity to contribute to your wonderful journal. Thank you very much.

Best regards,

Reviewer 3

Comments and Suggestions for Authors

The paper is a revision on the role of the oral microbiome as an indicator for systemic pathologies. The review is an extensive and cohomprehensive summary which sounds scientifically and is properly presented. However, a fundamental aspect have to be clarified: the paper is inteded to be a narrative review, but the authors used the terminology "in our systematic review", I please the authors to check for this aspect since the systematic review is a kind of paper which is so far from the author's one.

Response: The authors sincerely thank you for your precise and warm comments. As per your comment, this paper is a narrative review, not a systematic review. Therefore, we have corrected the corresponding term in manuscript.

The figures have to be inserted in the manuscript next to the citation point and not at the end of the paper, please check for authors guidelines.

Please, reference section have to be checked for discrepancies according to the authors guidelines.

Response: Thank you for your comment. This part you point out is technical, so we'll fix it with the journal. Thanks for pointing it out correctly.

Submission Date

21 June 2021

Date of this review

01 Jul 2021 11:08:11

Round 2

Reviewer 2 Report

Thank you for considering my comments